# Effects of Gadolinium Deposits in the Cerebellum: Reviewing the Literature from In Vitro Laboratory Studies to In Vivo Human Investigations

**DOI:** 10.3390/ijerph18147214

**Published:** 2021-07-06

**Authors:** Miski Aghnia Khairinisa, Winda Ariyani, Yoshito Tsushima, Noriyuki Koibuchi

**Affiliations:** 1Department of Integrative Physiology, Gunma University Graduate School of Medicine, Maebashi 371-8511, Japan; miskiaghniak@gmail.com; 2Program Study of Pharmacy, Faculty of Mathematics and Natural Sciences, Bandung Islamic University, Bandung 40116, Indonesia; 3Research Fellow of Japan Society for the Promotion of Science, Tokyo 102-0083, Japan; 4Department of Diagnostic Radiology and Nuclear Medicine, Gunma University Graduate School of Medicine, Maebashi 371-8511, Japan; yoshitotsushima@gunma-u.ac.jp; 5Gunma University Initiative for Advanced Research (GIAR), Maebashi 371-8511, Japan

**Keywords:** gadolinium, cerebellar development, neurotoxicity, thyroid hormone

## Abstract

Gadolinium (Gd)-based contrast agents (GBCAs) are chemicals injected intravenously during magnetic resonance imaging (MRI) to enhance the diagnostic yield. The repeated use of GBCAs can cause their deposition in the brain, including the cerebellum. Such deposition may affect various cell subsets in the brain and consequently cause behavioral alterations due to neurotoxicity. Caution should thus be exercised in using these agents, particularly in patients who are more likely to have repeated enhanced MRIs during their lifespan. Further studies are required to clarify the toxicity of GBCAs, and potential mechanisms causing neurotoxicity have recently been reported. This review introduces the effects of GBCAs in the cerebellum obtained from in vitro and in vivo studies and considers the possible mechanisms of neurotoxicity involved.

## 1. Introduction

Metal-containing contrast agents have been used to enhance magnetic resonance imaging (MRI). Among such metals, gadolinium (Gd) has been most commonly used. Gd is a metal that belongs to the lanthanide group. It becomes a trivalent cation with a +3 oxidation state. MRI images are mainly determined by the type of tissue and the pulse sequence for obtaining the images. A pulse sequence is a design of an image acquisition method consisting of transmitting a series of radio waves and receiving radio waves returning from the tissue. Hydrogen atoms (protons) of water and lipids contained in tissues have unique T1 (longitudinal direction) and T2 (transverse direction) relaxation times depending on the environment in which they are placed. The contrast of the image is mainly determined by T1 and T2 relaxation times. The images obtained by a method that emphasizes the difference in the T1 relaxation time of different tissues is called T1WI. Gd mainly has the effect of shortening the T1 relaxation time, so T1WI visualizes it as a high signal (white) [1,2]. As a contrast agent, Gd ion (Gd^3+^) contains the highest number of unpaired electrons, it has the properties of a magnet, which has the effect of shortening the T1 relaxation time of the surrounding protons [2]. However, because Gd^3+^ is highly toxic and disruptive to biological processes [3,4], it is chelated with an organic ligand molecule to produce highly soluble non-toxic complexes used as Gd-based contrast agents (GBCAs) [5]. GBCAs are classified as linear or macrocyclic, based on their chemical compositions, and ionic or non-ionic based on their ionic charges [6]. A recent study on the stability of GBCA revealed that 20% of linear GBCA may undergo transmetalation following 15 days of incubation [6]. Such transmetalation and the release of Gd^3+^ may cause toxicity in GBCA-deposited tissues, although the GBCA itself may also possibly cause toxicity.

Nine GBCAs are approved by the U.S. Food and Drug Administration (FDA) and have been used globally almost 100 million times. Due to the low sensitivity of MRI, large quantities of GBCAs must be injected into the patient to obtain useful images. However, concern over the safety of GBCAs, mostly linear GBCAs, regarding their deposition in issue has resulted in the restriction of its use by the European Medicine Agency and has triggered risk warnings from the FDA [7]. GBCAs are contraindicated in patients with severe kidney problems, in patients who are scheduled for or have recently received a liver transplant, and in newborn babies up to four weeks of age [https://apps.who.int (accessed on 30 June 2021)]. The deposition of GBCAs in various tissues including the brain has been reported [8,9]. Tissue deposits of linear GBCAs are much higher than those of macrocyclic GBCAs [10,11,12]. Among the brain regions, the dentate nucleus (DN) of the cerebellum has been considered a primary region for such deposition based on human studies [13,14]. Animal MRI studies have also revealed the same phenomenon [13,15]. Gd deposition has been observed in the DN and cerebellar cortex in model animals [16,17,18]. Gd compounds deposited in the cerebellum consist of both soluble and insoluble forms [7,19]. A large fraction of linear GBCAs is transformed to insoluble form and deposited in the cerebellum. Such deposition may affect various cell subsets and network circuits in the cerebellum. Although no evident histological changes in the brain have been found until now, repetitive injection of GBCAs has been shown to cause behavioral alterations in rodent models [17,18,20]. Such behavioral alterations may be induced by disrupting intracellular signaling or brain network connectivity by GBCA exposure without obvious changes in the cellular morphology.

The cerebellum is a central brain structure deeply integrated into primary loops with the cerebral cortex, brainstem, and spinal cord. It is known for its involvement in motor coordination such as in balance, the precise timing of movements, and motor learning [21,22]. In addition, recent brain imaging studies have shown the cerebellar contribution in cognitive functions such as attention, language, working memory, emotion, and visuospatial navigation [22]. The cerebellum is located in the posterior fossa and can be divided into three main regions: the flocculonodular lobe (anterior and inferior), the vermis (medial), and the two hemispheres (lateral) [21]. The cerebellum receives inputs from the cerebral cortex through the anterior pontine nuclei, several brainstem nuclei, and the spinal cord through the dorsal and ventral spinocerebellar tracts. Then, it emits outputs solely through the deep cerebellar nucleus (DCN), which projects to various brainstem nuclei and the cerebral cortex through the thalamus [21,22,23]. Cerebellar lesions may affect both cerebellar input and output, resulting in motor coordination disorder and ataxia.

Gd deposits in the cerebellum have raised concern regarding the safety and continued use of GBCAs. The adverse effects of GBCAs in the human cerebellum during development and adulthood remain uncertain due to several limitations, especially in the mechanisms of action and observation periods. GBCA deposition presents the potential risk of adverse effects on cerebellar function. Therefore, in vitro and in vivo animal models are necessary to elucidate the effects and mechanisms of GBCA exposure. This review explores the effects of GBCAs on the cerebellum based on the findings of in vitro and in vivo studies and discusses the possible mechanisms involved.

## 2. Gadolinium-Based Contrast Agent (GBCA) Deposition in the Cerebellum

As stated previously, the cerebellum is considered a primary region for GBCA deposition in the brain in humans and animals with repeat injections [13,14,15]. Increased signal intensity in the cerebellar DN on unenhanced T1-weighted images showed a positive correlation with previous GBCA exposure [24]. In animal studies, both single or repetitive injection of GBCAs resulted in the detachable presence of Gd in the cerebellar until five months after injection [10,11,25]. A human study examined Gd deposits in the DN, pons, globus pallidus (GP), and thalamus of 23 cadavers by electron microscopy, inductively coupled plasma mass spectrometry (ICP–MS), and light microscopy. Cadavers that received GBCAs before death had 0.1–58.8 µg Gd per gram of tissue in a significantly dose-dependent manner, which was consistent with changes of signal intensity on unenhanced T1W1. Gd accumulated in the brain in patients who received gadopentetate dimeglumine, gadodiamide, gadoteridol. They found detectable Gd deposits in the inner segment of the frontal lobe, cerebellar white matter, GP, and frontal lobe white matter. Furthermore, the DN and GP had the highest Gd deposition levels [9]. The same findings were also reported in pediatric patients who underwent repeated GBCA administration. A progressive hyperintensity was observed on unenhanced T1WI in the DN and GP [26,27]. However, no clear evidence has been established on whether the deposition is caused by chelated or de-chelated GBCAs.

Recent studies suggest that the deposition of Gd may be related to the stability of GBCAs, although the dissociation of GBCAs and subsequent release of free Gd^3+^ remain poorly understood [7,19]. The competitive binding of Gd^3+^ by endogenous anions such as PO_4_^3−^ and CO_3_^2−^ and transmetalation by endogenous metal cations including Zn^2+^, Fe^3+^, and Cu^2+^ are two possible demetalation pathways [28]. The order of affinity of GBCA chelators for endogenous cations is Fe^3+^ > Cu^2+^ > Zn^2+^. However, plasma iron is tightly regulated by transferrin, and serum Cu^2+^ concentrations are low (<10 µM). Thus, more attention has been paid to Zn^2+^ as a chelator because of its higher concentration in serum (up to 50 µM). Linear GBCA (i.e., Gd-DTPA and Gd-DTPA-BMA) undergo transmetalation with Zn^2+^ at pH > 4.5 [28]. On the other hand, previous mass spectrometry analysis has detected Fe-ligand complexes in the brain [28,29]. Furthermore, the addition of Gd^3+^ salts determines the Fe-ligand [6,29]. Although these results indicate the involvement of endogenous cations in transmetalation or Gd^3+^ release from GBCA, the roles of such cations on Gd deposition in the cerebellum have not yet been understood. Gd deposits in the cerebellum consist of both soluble and insoluble forms [7,19], indicating that Gd^3+^, at least in part, dissociated from GBCA and was subsequently deposited in cerebellar tissue. The insoluble form may consist of Gd^3+^ that was released from GBCA and bound with organic or inorganic anions, whereas soluble forms may consist of Gd-binding macromolecules and low–molecular weight Gd complexes which could be GBCAs that remained intact [7,19]. In any case, the exact chemical nature of the insoluble form has not yet been fully clarified [7,12]. Nevertheless, these results indicate that both Gd^3+^ and GBCAs may be deposited in the cerebellum and affect cerebellar function.

The in vivo animal model has been used to examine Gd deposition in several cerebellar regions. Transmission electron microscopy findings revealed that gadodiamide are deposited in the DCN, choroid plexus, and granular layer of the cerebellar cortex, whereas gadobenate are deposited in the DCN and choroid plexus [16]. Laser ablation ICP–MS showed the considerable deposition of linear GBCAs (gadopentetate, gadobenate, and gadodiamide) in the DCN and granular layer of the cerebellar cortex at concentrations of Gd ≥9.5 nmol/g, whereas macrocyclic GBCAs (gadobutrol, gadoterate, and gadoteridol) were found distributed throughout the cerebellum without any region-specific accumulation and at lower Gd concentrations (≤0.9 nmol/g), with some regions exhibiting a Gd concentration of at least 2.8 nmol/g [12]. These results likewise support the hypothesis that Gd deposition may affect various cell subsets in the cerebellum, including neurons and glial cells.

In addition to Gd deposition in adults, an animal study has shown that Gd can be deposited in the fetus if it is injected into a mouse mother [20]. This study has shown that repeated intravenous injection of linear (gadodiamide) or macrocyclic (gadoterate meglumine) GBCA from pregnancy day 15 to 19 induced deposition of Gd in the offspring brain at postnatal day 28. Greater levels of Gd were deposited several fold by linear GBCA injection than by macrocyclic GBCA injection and Gd were deposited in females in the greater levels than in males. In addition, a previous study has shown the transplacental transfer of GBCA in rabbits [30]. These results indicate that GBCA can cross the placenta and deposit in the brain. Due to the methodological limitations, however, the measurement was performed by using a whole brain. Thus, specific deposition in the cerebellum was not measured. However, by considering the studies in adult mice, it is reasonable to speculate that Gd can be deposited in the fetal cerebellum. The reason why greater levels of Gd were deposited by linear GBCA was not known. Such a difference could be induced by the difference of Gd^3+^ dissociation, or difference in level of GBCA crossing through the placenta. Furthermore, the reason why greater levels of Gd were deposited in females was also not known. However, sex-specific differences in the deposition of several toxic metals such as nickel, cadmium, and mercury have been reported [31], indicating that pharmacokinetics of toxic metals or contrast agents may differ between males and females. Further study is required to clarify the mechanisms causing such differences.

## 3. Effects of GBCAs on Cerebellar Structure and Function

### 3.1. Human Studies

A limited number of human studies have reported symptoms possibly caused by the cerebellar dysfunction due to GBCA deposition. One study found detectable Gd accumulation in the inner segment of the frontal lobe, cerebellar white matter, GP, and frontal lobe white matter. A study reported the possible clinical manifestations of 42 patients exposed to GBCAs, including central and peripheral pain, headache, bone pain, and skin changes [32]. Some such symptoms persisted beyond three months after injection. A report highlighted the onset of a series of symptoms; including neurological, musculoskeletal, dermal, and cognitive symptoms and balance disorders in patients within a month of their last MRI [33]. Another study showed that Gd retained in the brain may damage the GP and induce Parkinsonian symptoms [34]. Moreover, a retrospective study found that gadoterate administration in patients who had previously received more than 20 doses caused a clinical cerebellar syndrome [35]. However, due to a lack of retrospective or prospective clinical cohort studies regarding the association of post-exposure medical conditions and recurrent GBCA exposure, the clinical significance of cerebellar GBCA accumulation in humans during development and adulthood is not fully known. Nevertheless, because many studies have been published regarding cerebellar GBCA deposition in human children and adults, and because of the findings of animal model studies previously mentioned, the FDA has stated that GBCA administration should be considered during pregnancy only if it is essential and cannot be delayed [36].

Overall, the clinical consequences of Gd deposition in brain structures need to be explored in further studies prospectively evaluating motor, cognitive, and affective functions. Furthermore, the gadolinium deposition and toxicity mechanisms of Gd should also be elucidated to provide a basis for therapies for potential neurological deficits, especially cerebellar ataxia. The research community should take advantage of the clinically silent phase to speed up research and set up international registries. The myriad of cerebellar functions requires specific attention by the neurology community through dedicated studies.

### 3.2. Animal Studies

Animal studies have often been conducted to investigate the toxicity of Gd deposits in brain. Particularly, attentions have been paid on the Gd exposure during fetal or neonatal period. Previous studies in neonatal and juvenile rats have shown that intraventricular or intrathecal injection of GBCAs (gadopentetate dimeglumine and gadodiamide) induced behavioral and neurological alterations (i.e., focal seizure activity, ataxia, and delayed tremor) accompanied by histopathological changes (i.e., oligodendroglial loss, astrocytes hypertrophy, and eosinophilic granule formation) [37,38,39]. These neurological alterations may cause lower quality of life and persist throughout life.

Cerebellar neurotoxicity after GBCA exposure has been investigated both in vitro and in vivo (Table 1). In a mice study, intravenous administration of gadoterate meglumine (macrocyclic GBCA) or gadodiamide (linear GBCA) into dams during the prenatal period (embryonic days 15–19, a single injection/day), which is the critical period for the functional organization of neuronal circuits, disrupted motor coordination and impaired memory function, with a more severe phenotype observed in the gadodiamide-treatment group [20]. Motor coordination examined by rotarod was significantly disrupted in male and female mice. The disruption was more severe with linear GBCA than with macrocyclic GBCA, and more severe in females than in males. This difference may reflect the difference of Gd deposition: As discussed above, more Gd was deposited by linear GBCA injection than by macrocyclic GBCA and in females than in males, although the mechanisms causing such differences have not yet been clarified. Interestingly, although motor coordination was disrupted by prenatal GBCAs exposure, the motor learning ability was retained. In the adult rat, on the other hand, intracerebroventricular administration of gadopentetate dimeglumine (linear GBCA) impaired motor coordination [40]. Another study by the same group also showed that intracerebroventricular administration of gadodiamide caused acute and chronic ataxia, which were more severe than those treated by gadopentetate dimeglumine [37]. The exact brain regions affected by GBCA to create such behavioral abnormalities were not identified. Nonetheless, GBCAs likely mainly disrupt cerebellar function and impair motor coordination [41,42]. In the cerebellum, the accumulations of GBCAs have been reported as discussed above [43,44]. Morphological alteration of the cerebellar structure by GBCA treatment was also reported [37].

In spite of the fact that greater levels of Gd seem to be deposited in the cerebellum than in other brain regions, abnormal behaviors that may be caused by the disruption of other brain regions were also observed by prenatal GBCA injection [20]. For example, in the object recognition test (ORT), which is to test whether mice can recognize a novel object in an open field, GBCAs exposed mice showed lower recognition score than control animal, indicating impaired recognition memory. As for the rotarod test, the score was lower in linear GBCA-exposed mice. However, no sexual difference was observed. On the other hand, in the object-in-location test (OLT), which is to test the spatial memory, GBCAs exposed mice also showed lower score than control animals. As for the ORT, the score was lower with linear GBCA without sexual difference. The disruption of OLT indicates that mice cannot discriminate the object in a novel location due to suppressed memory acquisition. While the main brain region controlling ORT is the perirhinal cortex [45], the region controlling OLT is the hippocampus and entorhinal cortical [46], indicating that, in addition to the cerebellum, the function of other brain regions may be also affected by prenatal GBCA exposure. Unlike GBCAs’ action in the cerebellum, however, sexual difference was not observed. The mechanisms causing such differential GBCA action in other brain region have not yet been clarified. Studies to examine the toxicity in other brain region is also necessary.

The effects of GBCAs on cerebellar morphogenesis were studied further by in vitro studies [47]. Gadodiamide (linear GBCA) was found to exhibit an endocrine-disrupting effect by augmenting the thyroxine (T_4_)-activated dendritogenesis of Purkinje cells at low doses (10^−7^ M) and neurotoxic effects (decreased dendrite arborization and Purkinje cell survival) at higher doses (10^−5^ M). On the other hand, both GBCAs suppressed the triiodothyronine (T_3_)-activated dendritogenesis of Purkinje cell. Meanwhile, GdCl_3_ (most likely as Gd^3+^) showed severe neurotoxic effects even at low doses (10^−7^ M) and cell survival of cerebellar cells were greatly affected [47]. Not only Purkinje cells but also other types of cell were also affected by GBCA. Another in vitro study using a primary culture of cerebellar granule cells has shown alterations in granule cell survival by gadobutrol (macrocyclic GBCA) [48]. In addition, we are currently studying the effect of GBCAs on astrocyte function. Our preliminary study has shown the possible disruption of astrocyte function (manuscript in preparation). Taken together, GBCAs, particularly linear types, may affect motor coordination, at least in part, by affecting cerebellar morphogenesis or function.

## 4. Possible Mechanisms of GBCA Toxicity

Although the mechanism of GBCA toxicity has not yet been fully understood, their potential toxicity has been explored in vivo and in vitro. Some in vitro studies have shown that Gd toxicity is brought on by competitive inhibition of biological processes that require calcium [4,6,30,49]. Thus, most researchers believe that GBCA toxicity is induced by free Gd^3+^ that is released by dechelation. The ionic radius of Gd^3+^ is 107.8 pm, which is close to that of Ca^2+^ (114 pm) [50]. Such a similarity in size is one of the reasons for Gd^3+^ toxicity; it can compete with Ca^2+^ in all biological systems and in some enzyme activity that require Ca^2+^ but with a much higher binding affinity [29]. However, as mentioned previously, Gd occurs in insoluble and soluble forms. While the insoluble form may involve free Gd^3+^, the soluble form may be GBCAs, at least in part [7,12,19]. Gadodiamide, but not gadoterate meglumine, has been found to disrupt Ca^2+^ signaling in mouse GT1-7 neuronal cells that release gonadotropin-releasing hormone (GnRH) [51]. High doses (500 µmol/mL) of gadodiamide, gadobutrol, and gadobenate dimeglumine also reduced intracellular calcium levels in dorsal root ganglion neuronal cultures [5]. Both linear and macrocyclic GBCAs (gadodiamide, gadopentetate dimeglumine, gadoxetate disodium, gadobenate dimeglumine, gadoterate meglumine, gadobutrol, and gadoteridol) administered at 10 mM reduced mitochondrial function by reducing oxygen consumption and cell viability in human SH-SY5Y neuronal cells [52]. These results cannot be fully explained by the dissociation of Gd^3+^. Thus, other mechanisms may be involved in the GBCA-induced disruption of intracellular signaling pathways. Further studies to clarify such pathways are currently underway.

Some recent studies have provided novel information on GBCA toxicity in the cerebellum [16,20,43]. One study suggests that GBCAs disrupt the action of the thyroid hormone (TH), which plays a critical role in cerebellar development [53]. Disruption of TH action during critical period induces abnormal cerebellar development causing disruption motor coordination and learning. Purkinje cell dendrite arborization is retarded by perinatal hypothyroidism. Migration of granule cell from external granule cell layer to the internal granule cell layer was also retarded. Synaptogenesis between parallel fiber and Purkinje cell dendrite was also disrupted causing disrupted motor coordination and learning. TH action is mainly exerted by binding to the nuclear TH receptor (TR), although recent studies also showed that a part of TH action is also exerted though membrane receptor. TR binds to the TH response element (TRE) located in the promoter region of the target gene. Upon binding to TH, TR activates transcription of the target gene. TR is expressed all subset of cells throughout the brain including the cerebellum [53]. Considering the critical role of TH in cerebellar functional development, it is reasonable to speculate that GBCAs action may be at least in part caused by disrupting TH action in the cerebellum.

In a transient transfection-based reporter assay, using several different TREs gadodiamide, but not gadoterate meglumine, augmented TH receptor (TR)-mediated transcription at low doses (10^−7^ M) but suppressed it at high doses (10^−4^ M) [47]. GdCl_3_ did not exhibit such action. The same tendency was also demonstrated in T_4_-treated primary cerebellar cultures, in which a low dose (10^−7^ M) of gadodiamide augmented the dendrite arborization of Purkinje cells, whereas a high dose (10^−4^ M) suppressed it [47]. In T_3_-treated primary cerebellar cultures, however, both gadodiamide and gadoterate meglumine suppressed the dendrite arborization of Purkinje cells at 10^−5^ M (Figure 1) and the total Purkinje cell number [47]. Although the mode of action of GBCA on the TH system has not yet been fully clarified, GBCAs may disrupt calcium signaling by blocking the Ca^2+^ channel, which then affects the Ca^2+^/calmodulin-dependent protein kinase type IV (CaMKIV)-mediated augmentation of TR action [54] or binds to membrane receptor (i.e., integrin αvβ3) of known TH–binding sites [55]. Interplay between nuclear TR and membrane TH receptors may be possible, because GBCAs action seems to be different between T_4_ and T_3_ mediated dendritogenesis. Affinity to nuclear TR is greater in T_3_, whereas T_4_ has greater affinity to the membrane receptor and activates intracellular signal cascade. Thus, GBCAs may not only affect nuclear TR-mediated action but also affect membrane TH receptor-mediated action. Taken together with in vivo results, GBCAs, particularly the linear form, may cause toxic effects in the developing mouse brain, at least in part by disrupting the action of TH, thereby inducing behavioral alterations. Needless to say that GBCAs can also affect other various cellular signal cascades causing neurotoxicity. Further study is necessary to clarify further the neurotoxicity of GBCAs to select appropriate GBCA compound for safe brain imaging.

## 5. Conclusions

GBCAs can add tremendous value to MRI examinations by providing positive contrast, enhancing the image quality of lesions, and enabling dynamic imaging of vasculature and tissue perfusion over time. Nevertheless, the growing literature on the stability and cerebellar deposition of GBCAs as demonstrated by in vitro, in vivo, and human research data presents increasing evidence of potential toxicity associated with GBCA treatment (Figure 2). Such toxicity must be seriously and urgently considered. Further studies that provide new information on preventing GBCA-related toxicity, treating existing GBCA-related health issues, guiding the use of existing GBCAs, and directing the design of safer MRI contrast agents are essential. It should be noted that the FDA stated, “FDA warns that gadolinium-based contrast agents (GBCAs) are retained in the body; requires new class warnings (19 December 2017)”. Meanwhile, increased caution should be exercised in the repeated administration of GBCAs, particularly during pregnancy and lactation.

## Figures and Tables

**Figure 1 ijerph-18-07214-f001:**
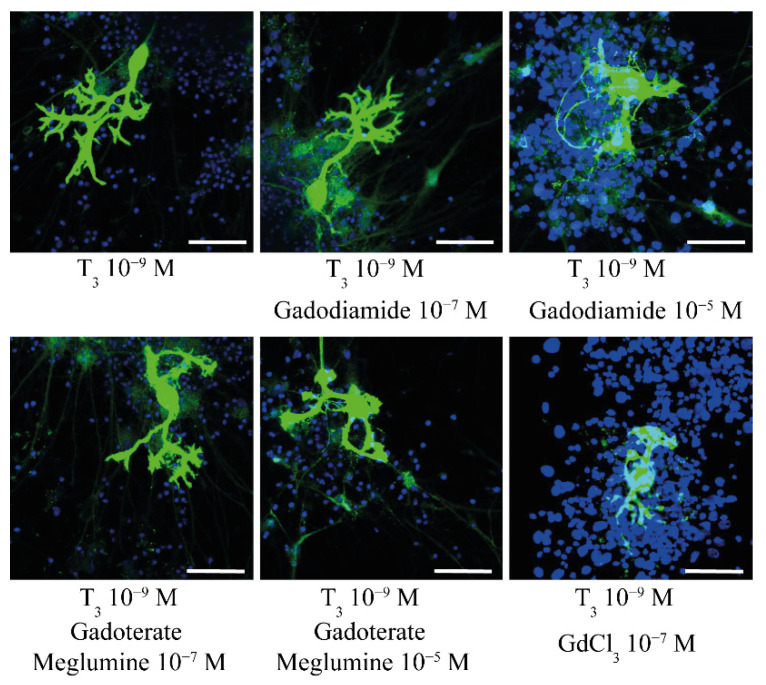
Representative photomicrographs showing the effects of gadodiamide (Gd−DTPA−BMA), gadoterate meglumine (Gd−DOTA), and GdCl_3_ on T_3_-mediated dendrite arborization of mouse Purkinje cells. Scale bar, 50 μm.

**Figure 2 ijerph-18-07214-f002:**
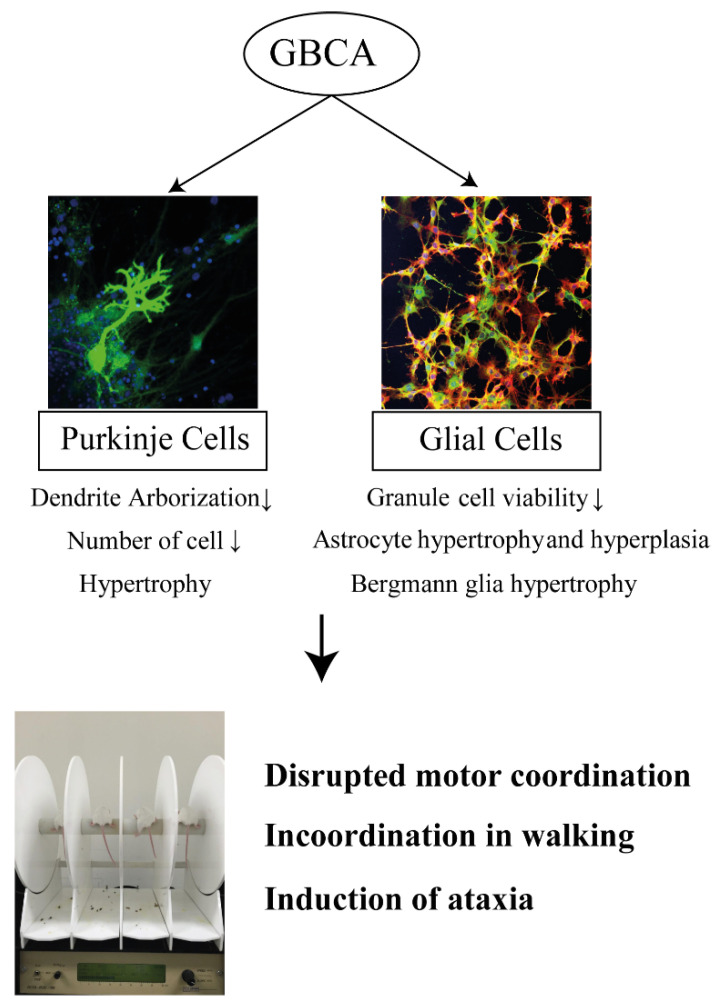
Possible mechanism of GBCA toxicity in the cerebellum.

**Table 1 ijerph-18-07214-t001:** Representative in vitro and in vivo studies investigating the cerebellar neurotoxicity of GBCAs.

Investigators	Model of Study	Type of GBCAs	Doses	Durations	Findings
Toczylowska et al., 2014 [48]	In Vitro: Primary culture of cerebellar granule cells	Gadobutrol	0.1, 1 and 10 mM	30–60 min exposure at DIV7	Reduction of the toxic effects of indocyanine green by increasing the cell viability of cerebellar granule cells culture 24 h after the exposure.
Ariyani et al., 2016 [47]	In Vitro: Primary culture of cerebellar Purkinje cells	GdCl3; gadodiamide; gadoterate meglumine	0.1 and 10 µM	DIV1–DIV17	Alteration of dendrite arborization and cell number of Purkinje cells by gadodiamide and GdCl_3_, but not gadoterate meglumine.
Ray et al., 1996 [38]	In Vivo: Rat	Gadopentetate dimeglumine	10 µmol/g brain (brain weight estimates at 2 g)	Single intracerebroventricular injection	Morphology: Focal lesions within the thalamus, brain stem, and spinal cord, with necrosis of glia, loss of myelin, and sparing of neurons and nerve fibers, but no description of cerebellar morphology. Behavior: Induction of ataxia with various grades starting at 28–220 min after injection, and lasting for up to 8 days.
Ray et al., 1998 [37]	In Vivo: Rat	gadodiamide; gadopentetate dimeglumine	10 µmol/g brain (brain weight estimates at 2 g)	Single intracerebroventricular injection	Morphology: Astrocyte hypertrophy and hyperplasia by gadodiamide, starting from 3 days after the injection. Hypertrophy of Bergmann glia and Purkinje cell Behavior: Incoordination in walking for several hours after gadodiamide injection. Development of ataxia and a high-stepping gait by a higher dose of gadodiamide later than 4 days after injection. Development of ataxia by gadopentetate dimeglumine at 24 h after injection; The symptom being more severe with transient anesthesia.
Khairinisa et al., 2017 [20]	In Vivo: Mouse	gadodiamide; gadoterate meglumine	2 mmol/kg body weight per day	day 15 to 19 of pregnancy	Induced of the disrupted motor coordination and impaired memory function by gadodiamide and gadoterate meglumine in adulthood.

GBCAs, gadolinium-based contrast agents; DIV, days in vitro.

## Data Availability

Data is available upon request.

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
