# Peer review of "Effects of Gadolinium Deposits in the Cerebellum: Reviewing the Literature from In Vitro Laboratory Studies to In Vivo Human Investigations"

_ijerph, 2021, doi:10.3390/ijerph18147214_

Round 1
Reviewer 1 Report
The manuscript entitled “Effects of Gadolinium deposits in the cerebellum: reviewing the literature from in vitro laboratory studies to in vivo human investigations” by Khairinisa et al is a very interesting work, addressing a topic that deserves more attention. The introduction is really informative and results are clearly exposed. Nevertheless, I have some questions with the aim to improve the manuscript.
Major comments:
1.- Please, explain the meaning for 1/T1 and 1/T2 (line 29).
2.- If MR is derived from relaxation of protons, please explain why to use Gd, what contains the highest number of unpaired electrons (line 34). How do you link protons and electrons?.
3.-It would be extremely interesting for clinicians to know the amount of Gd3+ administered usually in an clinical MR study. In this sense, readers would appreciate to have a relationship between this conventional Gd3+ supply during conventional an MR and toxic doses reported in the manuscript.
4- Define T1W1 (line 95).
5.- I can’t understand the sentence “The research community should take advantage of the clinically silent phase to speed up research” (lines 182-183). What silent phase and what kind of advantage should we expect?.
6.- The paragraph between lines 187-194, under the title of Animal studies, in fact mix data from animal models and humans. Please separate both kind of results.
7.- A figure to synthesize or summarize the postulated mechanisms of toxicity would be appreciated. This would be very important for clinicians and I think that the work would gain more readability.
Minor comments:
1.- Remove comma after the word “on” at line 40.
Reviewer 2 Report
- Please provide the explanation of the 1/T1 and 1/T2 in line 29.
- What is T1W1 in line 95
- It should be “an animal study has shown” in line 137.
- In line 145, there is a mistake “A previous study has shown the transplacental transfer of GBCA in 144 rabbits [31], indicating that GBCAs can also cross the placenta in mice”. Why does the rabbits study indicate the results in mice?
- General suggestion: please provide an abbreviation list in the manuscript since there are many abbreviations used in this review.
Reviewer 3 Report
Dear Editor
Despite the great increase in the resolution of MRI images, it is already known that GBCA traces accumulate in different tissues. Although the nature of their accumulation is not well understood, and even if the accumulated traces can actually bring further complications, the use of GBCAs has to be done with parsimony and care.
This review brings a collection of relevant information on the subject, focusing on the cerebellum, although often information on other brain structures appers in the text.
I consider the paper well organized and written
To complement the information on contraindication I ask the authors to consider inclusion in the introduction and conclusion
"GBCAs are contraindicated in patients with severe kidney problems, in patients who are scheduled for or have recently received a liver transplant, and in newborn babies up to four weeks of age." Pharmaceuticals: Restrictions in Use and Availability." World Health Organization. 2010. p. 14.
And in the conclusions emphasize the FDA's recommendations, "FDA warns that gadolinium-based contrast agents (GBCAs) are retained in the body; requires new class warnings." United States Food and Drug Administration. 2017-12-19.
Best regards
